# $\sigma$-ZERO: GRADIENT-BASED OPTIMIZATION OF $\ell_0$-NORM ADVERSARIAL EXAMPLES

## ABSTRACT

Evaluating the adversarial robustness of deep networks to gradient-based attacks is challenging. While most attacks focus $\ell_2$-norm and $\ell_\infty$-norm constraints to craft input perturbations, only a few have investigated sparse $\ell_1$-norm and $\ell_0$-norm attacks. In particular, $\ell_0$-norm attacks remain the least studied due to the inherent complexity of optimizing over a non-convex and non-differentiable constraint. However, evaluating the robustness of these attacks might unveil weaknesses otherwise left untested with conventional $\ell_2$ and $\ell_\infty$ attacks. In this work, we propose a novel $\ell_0$-norm attack, called $\sigma\text{-zero}$, which leverages an ad-hoc differentiable approximation of the $\ell_0$ norm to facilitate gradient-based optimization. Extensive evaluations on MNIST, CIFAR10, and ImageNet datasets, involving robust and non-robust models, show that $\sigma\text{-zero}$ can find minimum $\ell_0$-norm adversarial examples without requiring any time-consuming hyperparameter tuning, and that it outperforms all competing attacks in terms of success rate and scalability.

## 1 INTRODUCTION

Early research has unveiled that Deep Neural Networks (DNNs) are fooled by adversarial examples, i.e., slightly-perturbed inputs optimized to cause misclassifications (Biggio et al., 2013; Szegedy et al., 2014a; Goodfellow et al., 2015). In turn, this has demanded the need for more careful reliability assessments of such models. Most of the gradient-based attacks proposed to evaluate adversarial robustness of DNNs optimize adversarial examples under different $\ell_p$-norm constraints. In particular, while convex $\ell_1$, $\ell_2$, and $\ell_\infty$ norms have been widely studied (Chen et al., 2018; Croce & Hein, 2021a), only a few $\ell_0$-norm attacks have been considered so far. The main reason is that ad-hoc heuristics need to be adopted to compute efficient projections on the $\ell_0$ norm, overcoming issues related to its non-convexity and non-differentiability. Although this task is challenging and computationally expensive, attacks based on the $\ell_0$ norm have the potential to reveal uncovered issues in DNNs that may not be evident in other norm-based attacks (Carlini & Wagner, 2017; Croce & Hein, 2021a). For instance, these attacks, known for perturbing a minimal fraction of input features, can be used to determine the most sensitive characteristics that influence the model's decision-making process. Furthermore, they offer a different and relevant threat model to benchmark existing defenses. Developing efficient algorithms for generating $\ell_0$ adversarial examples is thus a crucial area of research that requires further exploration to improve current adversarial robustness evaluations.

Unfortunately, current implementations of $\ell_0$ attacks exhibit a largely suboptimal tradeoff between their success rate and efficiency, i.e., they are either accurate but slow, or fast but inaccurate. In particular, the accurate ones resort to the use of complex projections to find smaller input perturbations but suffer from time or memory limitations, hindering their scalability to larger networks or high-dimensional data (Brendel et al., 2019; Césaire et al., 2021). Other attacks execute faster, but their output solution is typically inaccurate and largely suboptimal as they rely on heuristic approaches and imprecise approximations to bypass the difficulties of optimizing the $\ell_0$ norm, leading to overestimating adversarial robustness (Matyasko & Chau, 2021; Pintor et al., 2021). However, all existing strategies are often slow to converge because they require a large number of queries (i.e., forward and backward passes), or they output suboptimal solutions. It thus remains an open challenge to develop a scalable and compelling method for assessing the robustness of DNNs against sparse perturbations with minimum $\ell_0$ norm.

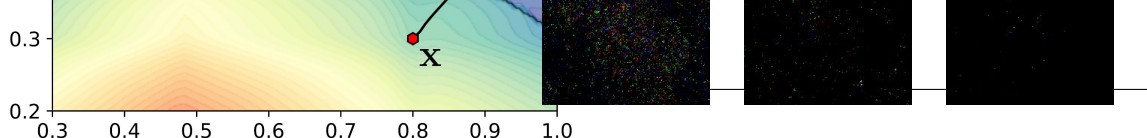

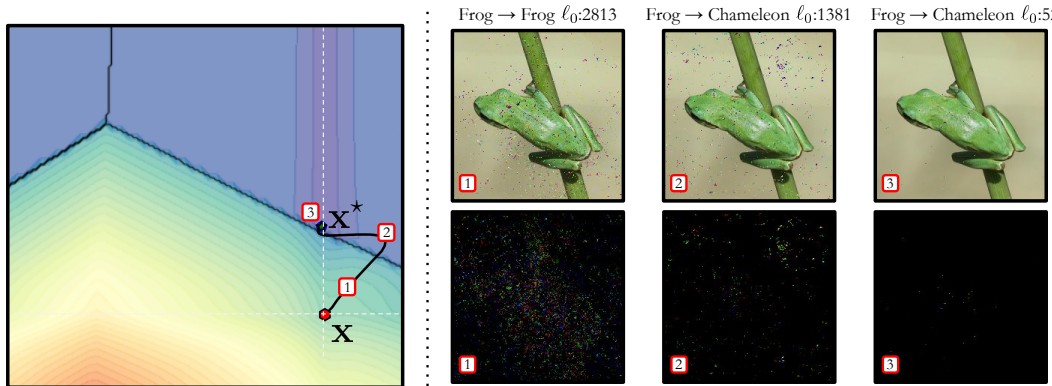

Figure 1: The leftmost plot shows an instance of $\sigma$-zero's execution on a two-dimensional problem. The red dot and the green star respectively represent the initial point $\mathbf{x}$ and the corresponding adversarial example $\mathbf{x}^\star$. Our gradient-based attack seeks to find this adversarial example while minimizing the number of perturbed features (i.e., the $\ell_0$ norm of the perturbation). Gray lines surrounding $\mathbf{x}$ demarcate regions where the $\ell_0$ norm is minimized. The rightmost plot shows the adversarial images (top row) and the corresponding perturbations (bottom row) found by $\sigma$-zero during the three steps highlighted in the leftmost plot, alongside their prediction and $\ell_0$ norm.

To tackle these issues, in this work we propose a novel attack technique, namely $\sigma$-zero, which iteratively promotes the sparsity of the adversarial perturbations, minimizing their $\ell_0$ norm (see Fig. 1 and Sect. 2). The underlying idea is to leverage a differentiable approximation of the actual $\ell_0$ norm, which is better suited to gradient-based optimizers. Specifically, we employ the approximation initially introduced by Osborne et al. (2000b), and more recently adopted by Cinà et al. (2022) for staging energy-latency poisoning attacks. This method offers an unbiased, differentiable estimation of the true $\ell_0$ norm, allowing us to optimize it via gradient descent.

Our experiments (Sect. 3) provide compelling evidence of the remarkable performance of our attack. We evaluate $\sigma$-zero on several benchmark datasets, including MNIST, CIFAR10, and ImageNet, considering baseline and robust models from Robustbench (Croce et al., 2021). We compare its performance with state-of-the-art attacks, showing that $\sigma$-zero achieves better results in terms of attack success rate and perturbation size, while being significantly faster and without requiring any sophisticated and time-consuming hyperparameter tuning. Overall, our approach encompasses two fundamental characteristics for a proficient adversarial attack, i.e., effectiveness and scalability, making it a catalyst for significant advancements in developing novel models with improved robustness, as well as better robustness evaluation tools.

## 2  $\sigma$-ZERO: MINIMUM $\ell_0$-NORM ADVERSARIAL EXAMPLES

We present here $\sigma$-zero, our gradient-based approach to finding minimum $\ell_0$-norm adversarial examples. We start by describing the considered threat model and then give a formal overview of the proposed attack and its algorithmic implementation.

**Threat Model.** We assume that the attacker has complete access to the target model, including its architecture and trained parameters, and exploits its gradient for staging white-box untargeted attacks. This setting is useful for worst-case evaluation of the adversarial robustness of DNN models, providing empirical upper bounds on the performance degradation that may incur when they are attacked, and it is the usual setting adopted also in previous work related to gradient-based adversarial robustness evaluations (Carlini & Wagner, 2017; Croce et al., 2021; Pintor et al., 2021).

**Problem Formulation.** In this work, we seek untargeted minimum $\ell_0$-norm adversarial perturbations that steer the model's decision towards misclassification. To this end, let $\mathbf{x} \in \mathcal{X} = [0,1]^d$ be a $d$-dimensional input sample, $y \in \mathcal{Y} = \{1, \ldots, l\}$ its associated true label, and $f : \mathcal{X} \times \Theta \mapsto \mathcal{Y}$ the target model, parameterized by $\theta \in \Theta$. While $f$ outputs the predicted label, we will also use $f_k$ to denote the continuous-valued output (logit) for class $k \in \mathcal{Y}$. The goal of our attack is to find the

minimum $\ell_0$-norm adversarial noise $\boldsymbol{\delta}^\star$ such that the corresponding adversarial example $\mathbf{x}^\star = \mathbf{x} + \boldsymbol{\delta}^\star$ is misclassified by $f$. This is formalized as the following optimization problem:

$$\boldsymbol{\delta}^\star \in \arg\min_{\boldsymbol{\delta}} \quad \|\boldsymbol{\delta}\|_0 \,, \tag{1}$$

$$\text{s.t.} \quad f(\mathbf{x} + \boldsymbol{\delta}, \boldsymbol{\theta}) \neq y \,, \tag{2}$$

$$\mathbf{x} + \boldsymbol{\delta} \in [0, 1]^d \,, \tag{3}$$

where $\| \cdot \|_0$ denotes the $\ell_0$ norm, which counts the number of non-zero dimensions. The hard-constraint in Equation 2 ensures that the perturbation $\boldsymbol{\delta}$ induces the target model $f$ to misclassify the perturbed sample $\mathbf{x}^\star$. Finally, Equation 3 represents a box constraint, ensuring that the adversarial example $\mathbf{x}^\star$ lies in $[0, 1]^d$. Note that when the source point $\mathbf{x}$ is already misclassified by $f$, the trivial solution to the above minimization problem is $\boldsymbol{\delta}^\star = \mathbf{0}$.

Contrary to the $\ell_1, \ell_2, \ell_\infty$ norms, when considering the $\ell_0$ norm the problem becomes intractable with standard methods. The $\ell_0$ norm is indeed non-differentiable, thus unsuitable for gradient-based optimization. To address this issue, we exploit the $\ell_0$-norm approximation function proposed by Osborne et al. (2000b), and defined as:

$$\hat{\ell}_0(\mathbf{x}) = \sum_{i=1}^{d} \frac{x_i^2}{x_i^2 + \sigma}, \qquad \sigma > 0, \qquad \hat{\ell}_0(\mathbf{x}) \in [0, d] \,, \tag{4}$$

with $\sigma$ being a hyperparameter controlling its approximation quality. When $\sigma$ tends to zero, the approximation becomes more accurate. However, an increasingly accurate approximation could lead to the same optimization limits of the $\ell_0$ norm.

Finally, similarly to previous work (Carlini & Wagner, 2017; Rony et al., 2021a; Szegedy et al., 2014b), we transform the hard-constraint in Equation 2 in a soft-constraint. The resulting optimization problem therefore becomes:

$$\boldsymbol{\delta}^\star \in \arg\min_{\boldsymbol{\delta}} \quad \mathcal{L}(\mathbf{x} + \boldsymbol{\delta}, y, \boldsymbol{\theta}) + \frac{1}{d}\hat{\ell}_0(\boldsymbol{\delta}) \tag{5}$$

$$\text{s.t.} \quad \mathbf{x} + \boldsymbol{\delta} \in [0, 1]^d \,, \tag{6}$$

where we substituted the $\|\boldsymbol{\delta}\|_0$ with the approximation $\hat{\ell}_0(\boldsymbol{\delta})$ and normalize it with respect to the number of features $d$ to ensure that its value is within the interval $[0, 1]$. The loss $\mathcal{L}$ is defined as:

$$\mathcal{L}(\mathbf{x}, y, \boldsymbol{\theta}) = \max\left( f_y(\mathbf{x}, \boldsymbol{\theta}) - \max_{k \neq y} f_k(\mathbf{x}, \boldsymbol{\theta}), 0 \right) + \mathbb{I}(f(\mathbf{x}, \boldsymbol{\theta}) = y) \,. \tag{7}$$

The first term in $\mathcal{L}$ represents the logit difference, which is positive when the sample is correctly assigned to the true class $y$, and clipped to zero when it is misclassified (Carlini & Wagner, 2017). The second term merely adds 1 to the loss if the sample is correctly classified.[1] This ensures that the loss term $\mathcal{L}$ is 0 only when an adversarial example is found, and higher than 1 otherwise. This in turn implies that the loss term $\mathcal{L}$ is always higher than the $\ell_0$-norm term in Equation 5 (as the latter is bounded in $[0, 1]$), when no adversarial example is found. Accordingly, it is not difficult to see that the feasible solutions of this problem only correspond to minimum-norm adversarial examples. It is also worth remarking that, conversely to the objective function proposed by Carlini & Wagner (2017), our objective does not require tuning the tradeoff between minimizing the loss and reducing the perturbation size to find minimum-norm adversarial examples, thereby avoiding a computationally-expensive line search for each input sample. In fact, the proposed objective function inherently induces an *alternate* optimization process between the loss term and the $\ell_0$-norm penalty, as shown in the Appendix (see Figure 4). In particular, when the sample is not adversarial, the attack algorithm mostly aims to decrease the loss term to find an adversarial example, while increasing the perturbation size. Conversely, when an adversarial example is found, the loss term is cropped to zero, and the perturbation size is gradually reduced.

**Solution Algorithm.** Given that the approximation function $\hat{\ell}_0$ in Equation 4 is differentiable, we derive a custom gradient-based algorithm for solving Equation 5 and Equation 6. Our attack, detailed

---

[1]While a sigmoid approximation may be adopted to overcome the non-differentiability of the $\mathbb{I}$ term at the decision boundary, we simply set its gradient to zero *everywhere*, without any impact on the experimental results.

---

**Algorithm 1:** $\sigma-\texttt{zero}$ Attack Pseudocode.

---

**Input:** $\mathbf{x} \in [0,1]^d$, input sample; y, true class label; $\boldsymbol{\theta}$, target model; N, number of iterations; $\quad\quad \sigma, \hat{\ell}_0$-approximation parameter; $\eta_0$, initial step size; $\tau_0$, initial sparsity threshold.

**Output:** $\mathbf{x}^{\star}$, minimum $\ell_0$ norm adversarial example.

1  $\boldsymbol{\delta} \leftarrow \mathbf{0}; \quad \boldsymbol{\delta}^{\star} \leftarrow \boldsymbol{\delta}; \quad \tau \leftarrow \tau_0; \quad \eta \leftarrow \eta_0$         ▷ initialization.
2  **for** $i$ in $1, \dots, N$ **do**
3      $\nabla\mathbf{g} \leftarrow \nabla_{\boldsymbol{\delta}}[\mathcal{L}(\mathbf{x} + \boldsymbol{\delta}, y, \boldsymbol{\theta}) + \frac{1}{d}\hat{\ell}_0(\boldsymbol{\delta}, \sigma)]$       ▷ gradient computation.
4      $\nabla\mathbf{g} \leftarrow \nabla\mathbf{g}/\|\nabla\mathbf{g}\|_{\infty}$         ▷ gradient normalization.
5      $\boldsymbol{\delta} \leftarrow \texttt{clip}(\mathbf{x} - [\boldsymbol{\delta} - \eta \cdot \nabla\mathbf{g}]) - \mathbf{x}$         ▷ $\boldsymbol{\delta}$ update.
6      $\boldsymbol{\delta} \leftarrow \Pi_{\tau}(\boldsymbol{\delta})$         ▷ zeroing $\boldsymbol{\delta}$ components below $\tau$.
7      $\eta = \texttt{cosine\_annealing}(\eta_0, i)$         ▷ $\eta$ update.
8      **If** $\mathcal{L}(\mathbf{x} + \boldsymbol{\delta}, y, \boldsymbol{\theta}) \leq 0 \; \tau+ = 0.01 \cdot \eta$ **else** $\tau- = 0.01 \cdot \eta$     ▷ $\tau$ update.
9  **end**
10 **If** $\mathcal{L}(\mathbf{x} + \boldsymbol{\delta}, y, \boldsymbol{\theta}) \leq 0 \;\wedge\; \|\boldsymbol{\delta}\|_0 < \|\boldsymbol{\delta}^{\star}\|_0 \;\; \boldsymbol{\delta}^{\star} \leftarrow \boldsymbol{\delta}$     ▷ $\boldsymbol{\delta}^{\star}$ update.
11 **return** $\mathbf{x}^{\star} \leftarrow \mathbf{x} + \boldsymbol{\delta}^{\star}$

---

in Algorithm 1, is fast, not memory-demanding, and easy to implement. It starts by initializing the adversarial perturbation $\boldsymbol{\delta} = \mathbf{0}$ (line 1). Subsequently, it computes the gradient of the objective function in Equation 5 with respect to $\boldsymbol{\delta}$ (line 3), and normalizes it to speed up convergence (Rony et al., 2018; Pintor et al., 2021). We then update $\boldsymbol{\delta}$ to minimize the objective via gradient descent, while also accounting for the box constraints in Equation 6 through the usage of the `clip` operator (line 5). We enforce sparsity in $\boldsymbol{\delta}$ by clipping to 0 all the components lower than the current sparsity threshold $\tau$ (Line 6). This step is necessary since the $\hat{\ell}_0$ approximation is not exact, and might result in some values being closer to zero but not precisely zero. We therefore encourage the attack to focus only on the most influential features, discarding less significant contributions. We then decrease the step size $\eta$ by following a cosine-annealing schedule (Rony et al., 2018; Pintor et al., 2021), and adjust the sparsity threshold $\tau$ dynamically. In particular, if the current sample is adversarial, we increase $\tau$ to promote sparser perturbations; otherwise, we decrease $\tau$ to reduce $\mathcal{L}$. The variations of $\tau$ are also iteratively reduced following the same cosine-annealing schedule of the step size. The above process is repeated for $N$ iterations, and if during each iteration, we find a better solution that is adversarial and has a lower $\ell_0$ norm, we update the optimal perturbation $\boldsymbol{\delta}^{\star}$ to the current minimum (line 10). Finally, the best adversarial perturbation $\boldsymbol{\delta}^{\star}$ identified during the optimization process is returned (line 11). In conclusion, the main contributions behind $\sigma-\texttt{zero}$ are: (i) the idea of exploiting the numerically-stable approximation of the $\ell_0$ norm by Osborne et al. (2000b) to design a novel loss function (Equation 5), which enables simultaneously searching for an adversarial example while minimizing the $\ell_0$ norm of the perturbation (i.e., a non-trivial task given the non-convexity of this norm); and (ii) the introduction of the sparsity threshold $\tau$ and its dynamic adjustment policy which, along with gradient normalization and step size annealing, help find very sparse adversarial perturbations faster. The combination of our novel formulation with the aforementioned optimization tricks yields a very fast and reliable $\ell_0$-norm attack algorithm, which does not even require specific hyperparameter tuning, as we will show in our experimental results.

## 3 EXPERIMENTS

We report the extensive evaluation of the proposed $\sigma-\texttt{zero}$ attack to compare its performance and efficiency with other state-of-the-art $\ell_0$ attacks, considering sixteen baseline and robust models and three different datasets.

### 3.1 EXPERIMENTAL SETUP

**Datasets.** We conduct experiments on three popular datasets used for benchmarking adversarial robustness: MNIST (LeCun & Cortes, 2005), CIFAR10 (Krizhevsky, 2009) and ImageNet (Krizhevsky et al., 2012). We use a random subset of 1000 test samples from ImageNet to evaluate attacks performance on it, while we consider the entire test set for MNIST and CIFAR10. For the MNIST and CIFAR10 datasets we used a batch size of 32, while for ImageNet we opted for a batch size of 16.

**Attacks.** We compare $\sigma$-zero against the following state-of-the-art, minimum-norm attacks, in their $\ell_0$-norm variants: the Voting Folded Gaussian Attack (VFGA) attack (Césaire et al., 2021), the Primal-Dual Proximal Gradient Descent (PDPGD) attack (Matyasko & Chau, 2021), the Brendel & Bethge (BB) attack (Brendel et al., 2019), including also its variant with adversarial initialization (BBadv), and the Fast Minimum Norm (FMN) attack (Pintor et al., 2021). We also consider two state-of-the-art $\ell_1$-norm attacks as additional baselines, i.e., the Elastic-Net (EAD) attack (Chen et al., 2018) and SparseFool (Modas et al., 2019), along with two further $\ell_0$-norm attacks, i.e., the $\ell_0$-norm Projected Gradient Descent (PGD-$\ell_0$) attack (Croce & Hein, 2019) and the Sparse Random Search (Sparse-RS) attack (Croce et al., 2022).[2] Compared to minimum-norm attacks, PGD-$\ell_0$ and Sparse-RS aim to maximize misclassification confidence within a given maximum number of modifiable features $k$. Thus, to ensure a fair comparison with minimum-norm attacks, as suggested by Rony et al. (2021b), we tune their perturbation budget $k$ by performing a sample-wise binary search to find minimum-norm adversarial examples. Further details are reported in the Appendix. Finally, we configure all attacks to manipulate input values separately, without constraining the manipulations to individual pixels; e.g., on CIFAR10, the number of modifiable inputs is thus $3 \times 32 \times 32 = 3072$.

**Models.** We use a selection of both baseline and robust models to evaluate the attacks under different conditions. Our goal is to compare $\sigma$-zero on a vast set of models to ensure its broad effectiveness and to expose vulnerabilities that may not be revealed by other attacks (Croce & Hein, 2021a). For the MNIST dataset, we consider two adversarially-trained convolutional neural network (CNN) models by Rony et al. (2021a), i.e., CNN-DDN and CNN-Trades. These models have been trained to be robust to both $\ell_2$ and $\ell_\infty$ adversarial attacks. We denote them respectively with M1 and M2. For the CIFAR10 and ImageNet datasets, we employ state-of-the-art robust models from RobustBench (Croce et al., 2021). For CIFAR10, we adopt eight models, denoted with C1-C10. C1 (Croce et al., 2021) is a non-robust WideResNet-28-10 model. C2 (Carmon et al., 2019) and C3 (Augustin et al., 2020) combine training data augmentation with adversarial training to improve robustness to $\ell_\infty$ and $\ell_2$ attacks. C4 (Engstrom et al., 2019) is an adversarially trained model that is robust to $\ell_2$-norm attacks. C5 (Gowal et al., 2021) exploits generative models to artificially augment the original training set and improve adversarial robustness to generic $\ell_p$-norm attacks. C6 (Chen et al., 2020) is a robust ensemble model. C7 (Xu et al., 2023) is a recently proposed adversarial training defense robust to $\ell_2$ attacks. C8 (Addepalli et al., 2022) enforces diversity during data augmentation and combines it with adversarial training. Finally, we also include the $\ell_1$ robust models C9 (Croce & Hein, 2021b) and C10 (Jiang et al., 2023). For ImageNet, we consider a pretrained ResNet-18 denoted with I1 (He et al., 2015), and five robust models to $\ell_\infty$-attacks, denoted with I2 (Engstrom et al., 2019), I3 (Wong et al., 2020), I4 (Salman et al., 2020), I5 (Hendrycks et al., 2021), and I6 (Salman et al., 2020).

**Hyperparameters.** We conduct our experiments using the default hyperparameters used in the original implementation of the attacks from AdversarialLib (Rony & Ben Ayed) and Foolbox (Rauber et al., 2017). We only change the number of steps to 1000, to ensure that all attacks reach convergence (Pintor et al., 2022). VFGA (Césaire et al., 2021) constitutes the only exception, as it terminates only once an adversarial example is obtained. We report additional results using 100 steps in the Appendix. As gradient-based attacks perform one forward and one backward pass in each step, we double the steps for Sparse-RS, which, being a gradient-free attack, only makes one forward pass per iteration. This ensures a fair comparison. For $\sigma$-zero, we set 1000 steps, $\eta_0 = 1$, $\tau_0 = 0.5$ and $\sigma = 0.1$. We keep the same configuration for all models and datasets, showing that no specific hyperparameter tuning is required for $\sigma$-zero. Additional analyses of the influence of the hyperparameters on the performance of $\sigma$-zero can be found in the Appendix.

**Evaluation Metrics.** For each attack, we report the Attack Success Rate (ASR), defined as the ratio of successfully attacked samples, and the median $\ell_0$ norm. Additionally, we report $ASR_k$, which indicates the ASR of attacks with a fixed budget of $k$ perturbed features. We also compare the computational effort of each attack considering their execution time, the average number of queries (i.e., the sum of #forwards and #backwards) needed to perform each attack, and the Video Random Access Memory (VRAM) consumption.[3] We measure the execution time on a workstation with NVIDIA A100 Tensor Core GPU (40GB memory) and two Intel® Xeo® Gold 6238R processors. For measuring the memory consumption, we consider the maximum amount of VRAM used by each

---

[2]Sparse-RS is a gradient-free (black-box) attack, which only requires query access to the target model. We consider it as an additional baseline in our experiments, but it should not be considered a direct competitor of gradient-based attacks, as it works under much stricter assumptions (i.e., no access to input gradients).

[3]VRAM is a type of memory designed explicitly for use in Graphics Processing Units (GPUs).

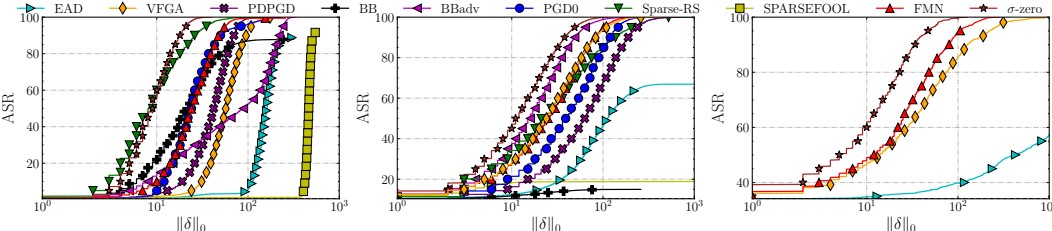

Figure 2: Robustness evaluation curves, reporting ASR versus perturbation size, for M2 on MNIST (leftmost plot) , C2 on CIFAR10 (middle plot), and I1 on ImageNet (rightmost plot).

attack among all the batches, which is a minimum requirement to run it without failure. By assessing the performance of each attack across these various metrics, we can gain a more comprehensive understanding of their effectiveness and scalability.

## 3.2 EXPERIMENTAL RESULTS

**Attack Performance.** Table 1 reports, for all models and datasets, the median value of $||\boldsymbol{\delta}||_0$ and the attack success rates. The values obtained confirm that our attack can find smaller perturbations in all cases. Specifically, over all the dataset-model configurations, $\sigma$-zero drastically improves the state of the art of sparse attacks. For example, on CIFAR10 models, $\sigma$-zero outperforms FMN by reducing the median number of manipulated features from 52 to 32 in the best case (C9) and from 7 to 5 in the worst case (C1). On ImageNet models, the median $||\boldsymbol{\delta}||_0$ is reduced from 58 to 23 in the best case (I6) and from 9 to 3 in the worse case (I2). Furthermore, we observe that the ASR of BB, which is the closest attack in terms of performance to $\sigma$-zero, drops when used in settings where the input dimensionality increases (e.g., CIFAR10), and it becomes unfeasible in extreme cases (i.e., ImageNet). From Table 1, we can also notice that the median $||\boldsymbol{\delta}||_0$ of BB sometimes is $\infty$, since its ASR is lower than $50\%$. BBadv does not suffer from the same issue but $\sigma$-zero continues to outperform that variant too. Lastly, we show in the Appendix that our attack always reaches ASR=100% against all models, even when decreasing the number of iterations. For other attacks, this is not ensured, particularly when reducing the number of iterations.

**Computational Effort.** We report the runtime comparison, the number of queries issued to the model, and the VRAM used by each attack. The results show that our attack is up to 2 (16) times faster than BB when considering MNIST (CIFAR10) models. Therefore, even if BB finds slightly better $\ell_0$-adversarial examples in one configuration, its computational effort is much higher than $\sigma$-zero. Furthermore, we observed that BB often stops unexpectedly before reaching the specified number of steps because it fails to initialize the attack.
The speed advantage of $\sigma$-zero is given because our attack is a simple gradient-based approach that avoids costly inner projections, such as the ones used by BB. On the other hand, $\sigma$-zero is slightly slower than FMN and VFGA; however, it compensates by finding better solutions. Notably, similarly to them, $\sigma$-zero requires fewer queries than remaining attacks. Furthermore, the speed-competing method VFGA is memory-hungry, forcing us to reduce the batch size when testing its effectiveness on larger models, e.g., C5, C6, and C7. Conversely, running our algorithm also requires reasonable VRAM, as $\sigma$-zero implements a lightweight search that includes only the cost of computing gradients and norms for each step. Overall, the practical advantages of our attack make it a promising direction for benchmarking large DNNs in an effective and time-efficient way.

**ImageNet Results.** For ImageNet, we restrict our analysis to EAD, FMN, and VFGA, as they outperform competing attacks on MNIST and CIFAR10 in terms of ASR, perturbation size, and execution time. While all ImageNet models are deemed robust to $\ell_1$ and $\ell_\infty$-norm attacks, they are vulnerable to our $\ell_0$-attack. Remarkably, I6 offers higher robustness against $\ell_0$ attacks, requiring more effort to evade it. The results show that in most configurations, our attack finds adversarial perturbations with a lower median $\ell_0$-norm, while being at the same time faster and memory-comparable. The results in the Appendix further confirm that even when decreasing the number of iterations to 100, our attack finds lower $\ell_0$-norm solutions and always achieves ASR=100%.

Table 1: Attack results with 1000 steps. For each attack, we report the corresponding $ASR_{10}$, $ASR_{50}$, ASR, median $\|\delta\|_0$, sample-level average execution time and the number of queries q (x1000), and the maximum VRAM consumed during the execution. The symbol * in VFGA suggests a potential overestimation due to using a smaller batch size for memory constraints.

| | | Performance | | | | Computational effort | | | | Performance | | | | Computational effort | | |
|---|---|---|---|---|---|---|---|---|---|---|---|---|---|---|---|---|---|
| Attack | Model | $ASR(\%)_{10}$ | $ASR(\%)_{50}$ | $ASR(\%)$ | $\|\delta\|_0$ | t(s) | q | VRAM | Model | $ASR(\%)_{10}$ | $ASR(\%)_{50}$ | $ASR(\%)$ | $\|\delta\|_0$ | t(s) | q | VRAM |
| | | | | | | | MNIST | | | | | | | | | |
| EAD | M1 | 1.14 | 53.66 | 100.0 | 49 | 0.47 | 6.28 | 0.05 | M2 | 1.2 | 55.57 | 100.0 | 48 | 0.5 | 6.73 | 0.05 |
| VFGA | | 9.62 | 82.42 | 99.98 | 27 | 0.05 | 0.77 | 0.21 | | 1.8 | 39.33 | 99.95 | 57 | 0.05 | 1.33 | 0.21 |
| PDPGD | | 2.97 | 74.08 | 100.0 | 38 | 0.23 | 2.0 | 0.04 | | 3.31 | 66.3 | 100.0 | 42 | 0.23 | 2.0 | 0.04 |
| BB | | 14.97 | 97.86 | 100.0 | 18 | 0.90 | 2.99 | 0.05 | | 25.95 | 91.62 | 100.0 | 18 | 0.74 | 3.71 | 0.05 |
| BBadv | | 14.81 | 91.23 | 100.0 | 19 | 0.77 | 2.01 | 0.07 | | 14.42 | 40.88 | 100.0 | 89 | 0.71 | 2.01 | 0.07 |
| PGD-$\ell_0$ | | 12.22 | 99.84 | 100.0 | 19 | 1.15 | 1.99 | 0.07 | | 5.04 | 90.17 | 100.0 | 24 | 1.42 | 2.0 | 0.06 |
| Sparse-RS | | 12.57 | 83.74 | 100.0 | 25 | 3.45 | 3.05 | 0.07 | | 60.04 | 98.48 | 100.0 | 9.0 | 2.51 | 2.44 | 0.06 |
| SPARSEFOOL | | 4.86 | 6.76 | 96.98 | 469 | 1.07 | 0.18 | 0.06 | | 0.93 | 1.21 | 91.68 | 463 | 2.87 | 0.86 | 0.07 |
| FMN | | 7.29 | 93.74 | 100.0 | 29 | 0.21 | 2.0 | 0.04 | | 10.86 | 91.84 | 99.41 | 24 | 0.22 | 2.0 | 0.04 |
| σ-zero | | 19.6 | **99.98** | 100.0 | **16** | 0.31 | 2.0 | 0.04 | | **61.57** | **100.0** | 100.0 | **9.0** | 0.31 | 2.0 | 0.04 |
| | | | | | | | CIFAR10 | | | | | | | | | |
| EAD | C1 | 6.74 | 21.33 | 100.0 | 126 | 2.32 | 6.9 | 1.47 | C5 | 14.47 | 35.9 | 100.0 | 74 | 10.76 | 5.55 | 9.92 |
| VFGA | | 48.58 | 93.41 | 99.99 | 11 | 0.17 | 0.32 | 11.96 | | 27.71 | 67.51 | 99.88 | 29 | 4.91* | 1.02 | >40 |
| PDPGD | | 16.58 | 78.97 | 100.0 | 27 | 0.64 | 2.0 | 1.31 | | 17.68 | 40.89 | 100.0 | 69 | 3.96 | 2.0 | 8.86 |
| BB | | **69.77** | 99.79 | 100.0 | 7 | 5.81 | 2.76 | 1.47 | | 13.46 | 17.14 | 17.94 | ∞ | 3.46 | 2.08 | 9.93 |
| BBadv | | 69.58 | 99.86 | 100.0 | 7 | 4.57 | 2.01 | 1.63 | | – | – | – | – | – | – | – |
| PGD-$\ell_0$ | | 31.82 | 85.21 | 100 | 18 | 6.46 | 1.93 | 1.73 | | – | – | – | – | – | – | – |
| Sparse-RS | | 59.64 | 98.59 | 100 | 9 | 3.76 | 1.49 | 1.74 | | – | – | – | – | – | – | – |
| SPARSEFOOL | | 11.19 | 11.19 | 56.56 | 3072 | 1.42 | 0.37 | 1.57 | | – | – | – | – | – | – | – |
| FMN | | 67.52 | 99.97 | 100.0 | 7 | 0.60 | 2.0 | 1.3 | | 27.97 | 68.38 | 100.0 | 29 | 3.91 | 2.0 | 8.86 |
| σ-zero | | 80.84 | **100.0** | 100.0 | **5** | 0.83 | 2.0 | 1.51 | | **44.72** | **94.14** | 100.0 | **12** | 4.39 | 2.0 | 9.92 |
| EAD | C2 | 12.7 | 30.38 | 100.0 | 90 | 1.92 | 5.70 | 1.47 | C6 | 17.29 | 33.68 | 100.0 | 105 | 8.33 | 5.37 | 5.39 |
| VFGA | | 28.98 | 75.37 | 99.92 | 24 | 0.59 | 0.78 | 11.71 | | 34.17 | 81.79 | 99.89 | 20 | 4.30* | 0.62 | >40 |
| PDPGD | | 16.47 | 42.50 | 100.0 | 63 | 0.64 | 2.0 | 1.32 | | 21.37 | 48.96 | 99.82 | 51 | 2.15 | 2.0 | 5.12 |
| BB | | 11.73 | 14.24 | 14.7 | ∞ | 0.63 | 2.05 | 1.47 | | 37.98 | 78.76 | 83.58 | 16 | 12.49 | 3.14 | 5.39 |
| BBadv | | 37.64 | 90.57 | 100 | 16 | 4.68 | 2.01 | 1.64 | | – | – | – | – | – | – | – |
| PGD-$\ell_0$ | | 21.4 | 56.85 | 100 | 39 | 5.79 | 1.92 | 1.75 | | – | – | – | – | – | – | – |
| Sparse-RS | | 31.02 | 62.81 | 90.78 | 27 | 6.6 | 1.89 | 1.71 | | – | – | – | – | – | – | – |
| SPARSEFOOL | | 18.31 | 18.77 | 56.39 | 3072 | 11.31 | 1.4 | 1.62 | | – | – | – | – | – | – | – |
| FMN | | 28.43 | 74.7 | 100.0 | 26 | 0.59 | 2.0 | 1.31 | | 33.3 | 79.7 | 100.0 | 21 | 2.05 | 2.0 | 5.12 |
| σ-zero | | **47.15** | **95.38** | 100.0 | **11** | 0.82 | 2.0 | 1.53 | | **49.29** | **97.14** | 100.0 | **11** | 2.71 | 2.0 | 5.39 |
| EAD | C3 | 9.21 | 11.42 | 100.0 | 360 | 2.53 | 5.62 | 1.89 | C7 | 9.38 | 23.62 | 100.0 | 148 | 2.23 | 5.8 | 2.15 |
| VFGA | | 21.82 | 66.5 | 99.62 | 34 | 0.48 | 0.94 | 16.53 | | 22.79 | 56.72 | 99.81 | 39 | 3.15* | 1.84 | >40 |
| PDPGD | | 13.96 | 45.15 | 100.0 | 55 | 1.12 | 2.0 | 1.8 | | 11.97 | 38.41 | 100.0 | 69 | 0.76 | 2.0 | 2.0 |
| BB | | 21.32 | 56.78 | 58.64 | 33 | 2.31 | 2.89 | 1.89 | | 38.82 | 93.24 | 100.0 | 15 | 6.49 | 2.87 | 2.16 |
| BBadv | | 31.64 | 96.31 | 100.0 | 17 | 3.92 | 2.01 | 1.99 | | – | – | – | – | – | – | – |
| PGD-$\ell_0$ | | 17.33 | 58.82 | 100.0 | 39 | 10.31 | 1.93 | 2.30 | | – | – | – | – | – | – | – |
| Sparse-RS | | 21.41 | 61.00 | 100.0 | 36 | 5.54 | 2.26 | 2.20 | | – | – | – | – | – | – | – |
| SPARSEFOOL | | 14.3 | 21.22 | 98.74 | 3070 | 3.62 | 0.46 | 1.90 | | – | – | – | – | – | – | – |
| FMN | | 20.61 | 71.7 | 100.0 | 33 | 1.08 | 2.0 | 1.8 | | 23.95 | 70.24 | 100.0 | 30 | 0.73 | 2.0 | 2.0 |
| σ-zero | | **36.61** | **97.55** | 100.0 | **15** | 1.41 | 2.0 | 1.92 | | **44.53** | **96.77** | 100.0 | **12** | 0.93 | 2.0 | 2.15 |
| EAD | C4 | 9.48 | 11.14 | 100.0 | 398 | 2.57 | 5.66 | 1.89 | C8 | 15.75 | 29.23 | 100.0 | 118 | 1.01 | 5.32 | 0.41 |
| VFGA | | 30.5 | 90.04 | 99.88 | 19 | 0.28 | 0.52 | 16.53 | | 29.55 | 74.15 | 99.54 | 25 | 0.17 | 0.77 | 3.07 |
| PDPGD | | 15.5 | 49.19 | 100.0 | 51 | 1.16 | 2.0 | 1.8 | | 19.43 | 41.0 | 100.0 | 66 | 0.44 | 2.0 | 0.36 |
| BB | | 16.32 | 31.03 | 31.36 | ∞ | 3.01 | 2.37 | 1.89 | | 38.64 | 91.83 | 100.0 | 15 | 10.90 | 2.93 | 0.41 |
| BBadv | | 37.06 | 99.11 | 100.0 | 14 | 4.51 | 2.01 | 1.99 | | 38.01 | 93.04 | 100.0 | 16 | 4.6 | 2.01 | 0.54 |
| PGD-$\ell_0$ | | 19.9 | 70.04 | 100.0 | 33 | 8.97 | 1.93 | 2.30 | | 24.20 | 59.98 | 100 | 36 | 4.1 | 1.9 | 0.56 |
| Sparse-RS | | 22.82 | 62.18 | 100 | 36 | 13.2 | 2.26 | 2.20 | | 31.51 | 67.82 | 98.46 | 27 | 9.84 | 3.95 | 0.54 |
| SPARSEFOOL | | 15.52 | 40.86 | 93.82 | 3039 | 9.3 | 1.56 | 1.90 | | 23.18 | 26.54 | 51.80 | 3072 | 0.58 | 0.33 | 0.51 |
| FMN | | 26.85 | 85.6 | 100.0 | 23 | 1.09 | 2.0 | 1.8 | | 29.75 | 73.71 | 100.0 | 16 | 0.41 | 2.0 | 0.36 |
| σ-zero | | **42.96** | **99.15** | 100 | **12** | 1.39 | 2.0 | 1.91 | | **44.29** | **94.21** | 100.0 | **13** | 0.63 | 2.0 | 0.51 |
| EAD | C9 | 12.96 | 13.23 | 100.0 | 800 | 0.94 | 4.89 | 0.65 | C10 | 23.94 | 24.78 | 100.0 | 768 | 1.04 | 4.99 | 0.65 |
| VFGA | | 18.86 | 49.98 | 99.72 | 51 | 0.32 | 1.25 | 4.44 | | 33.61 | 69.47 | 99.83 | 28 | 0.25 | 0.82 | 4.22 |
| PDPGD | | 15.95 | 35.13 | 100.0 | 75 | 0.41 | 2.0 | 0.59 | | 26.89 | 42.38 | 100 | 66 | 0.4 | 2.0 | 0.60 |
| BB | | 14.13 | 22.91 | 27.64 | ∞ | 1.04 | 2.25 | 0.65 | | 24.72 | 27.98 | 29.50 | ∞ | 0.54 | 2.09 | 0.65 |
| BBadv | | 19.93 | 72.43 | 100 | 34 | 5.28 | 2.01 | 0.64 | | 35.67 | 82.46 | 100 | 22 | 3.03 | 2.01 | 0.65 |
| PGD-$\ell_0$ | | 17.05 | 36.85 | 100.0 | 72 | 4.45 | 1.92 | 0.72 | | 28.2 | 45.42 | 100.0 | 60 | 4.44 | 1.85 | 0.70 |
| Sparse-RS | | 17.89 | 34.56 | 92.91 | 90 | 13.62 | 2.42 | 0.69 | | 30.61 | 48.57 | 95.45 | 54 | 6.29 | 2.03 | 0.68 |
| SPARSEFOOL | | 15.89 | 24.36 | 58.29 | 3072 | 1.63 | 0.48 | 0.66 | | 26.85 | 43.07 | 91.14 | 69 | 4.32 | 1.49 | 0.66 |
| FMN | | 18.61 | 48.87 | 100 | 52 | 0.24 | 2.0 | 0.60 | | 32.63 | 62.96 | 100.0 | 34 | 0.35 | 2.0 | 0.59 |
| σ-zero | | **21.49** | **73.02** | **100.0** | **32** | 0.43 | 2.0 | 0.71 | | **37.27** | **82.92** | 100.0 | **20** | 0.42 | 2.0 | 0.72 |
| | | | | | | | ImageNet | | | | | | | | | |
| EAD | I1 | 34.4 | 36.3 | 100.0 | 460 | 1.69 | 6.06 | 1.21 | I4 | 56.2 | 61.4 | 100.0 | 0 | 1.41 | 5.29 | 1.21 |
| VFGA | | 48.4 | 72.4 | 99.2 | 14 | 3.03* | 2.0 | >40 | | 61.6 | 76.6 | 99.3 | 1 | 3.44* | 1.21 | >40 |
| FMN | | 48.7 | 81.0 | 100.0 | 12 | 0.62 | 2.0 | 1.14 | | 63.8 | 78.7 | 100.0 | 0 | 0.57 | 2.0 | 1.14 |
| σ-zero | | **62.0** | **95.9** | 100.0 | **5** | 0.81 | 2.0 | 1.19 | | **75.5** | **92.8** | 100.0 | 0 | 0.72 | 2.0 | 1.27 |
| EAD | I2 | 44.6 | 51.0 | 100.0 | 42 | 3.64 | 5.67 | 4.36 | I5 | 26.5 | 28.4 | 100.0 | 981 | 3.53 | 5.49 | 4.36 |
| VFGA | | 49.1 | 63.4 | 96.7 | 12 | 15.17* | 2.35 | >40 | | 36.6 | 59.5 | 97.9 | 31 | 12.12* | 1.98 | >40 |
| FMN | | 50.9 | 67.0 | 100.0 | 9 | 1.21 | 2.0 | 4.25 | | 38.1 | 67.7 | 100.0 | 25 | 1.23 | 2.0 | 4.25 |
| σ-zero | | **63.1** | **87.4** | 100.0 | **3** | 1.43 | 2.0 | 4.43 | | **46.6** | **86.9** | 100.0 | **13** | 1.51 | 2.0 | 4.43 |
| EAD | I3 | 55.1 | 60.2 | 100.0 | 0 | 3.53 | 5.5 | 4.36 | I6 | 32.3 | 33.5 | 100.0 | 572 | 8.34 | 5.34 | 5.67 |
| VFGA | | 62.2 | 76.2 | 98.8 | 1 | 10.12* | 1.43 | >40 | | 35.4 | 46.5 | 95.5 | 66 | 52.32* | 3.95 | >40 |
| FMN | | 64.1 | 79.5 | 100.0 | 0 | 1.22 | 2.0 | 4.25 | | 35.6 | 47.2 | 100.0 | 58 | 3.15 | 2.0 | 5.54 |
| σ-zero | | **75.5** | **91.4** | 100 | 0 | 1.44 | 2.0 | 4.43 | | **40.7** | **65.1** | 100.0 | **23** | 3.75 | 2.0 | 5.91 |

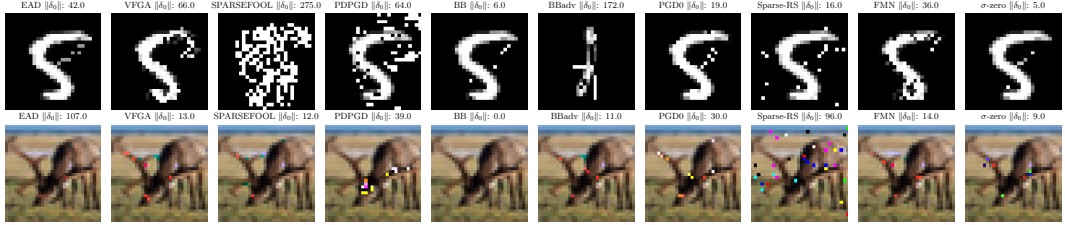

Figure 3: Randomly chosen adversarial examples from MNIST M2 (top-row), CIFAR10 C2 (bottom row) found by adversarial attacks we tested.

**Robustness Evaluation Curves.** Complementary to the performance results shown in Table 1, we present the robustness evaluation curves in Fig. 2 for each attack on M2, C2, and I1. These curves go beyond the only median statistic and $\text{ASR}_k$, providing more evidence that $\sigma\text{-zero}$ achieves higher ASR with smaller $\ell_0$-norm perturbations compared to the other attacks. Moreover, the ASR of our attack goes up to $100\%$, validating the correctness of our gradient-based approach even when considering unbounded perturbations (Carlini & Wagner, 2017). These results reinforce our previous findings that $\sigma\text{-zero}$ is an efficient and effective method for generating adversarial examples with smaller $\ell_0$ norm. In the Appendix, we include similar curves for all the other experimental configurations, for which results are consistent. In summary, our $\sigma\text{-zero}$ attack consistently outperforms other state-of-the-art methods, suggesting that it can identify smaller and more effective perturbations, making it a highly promising robustness evaluation method.

**Visual Inspection of Adversarial Examples.** In Fig. 3, we show adversarial examples generated with competing $\ell_0$-attacks, and our $\sigma\text{-zero}$. First, we can see that $\ell_0$ adversarial perturbations are always clearly visually distinguishable. Their goal, indeed, is not to be indistinguishable to the human eye – a common misconception related to adversarial examples (Biggio & Roli, 2018; Gilmer et al., 2018) – but rather to show whether and to what extent models can be fooled by just changing a few input values. For example, note how FMN and VFGA find similar perturbations, as they mostly target overlapping regions of interest. Conversely, EAD finds sparse perturbations scattered throughout the image but with a lower magnitude. This divergence is attributed to EAD's reliance on an $\ell_1$ regularizer, which promotes sparsity, thus diminishing perturbation magnitude without necessarily reducing the number of perturbed features. Conversely, our attack does not focus on specific areas or patterns within the images but identifies diverse critical features, whose manipulation is sufficient to mislead the target models. Given the diversity of solutions that the attacks offer, we argue that their combined usage may still improve adversarial robustness evaluation to sparse attacks.

## 4 RELATED WORK

Due to the inherent complexity of optimizing over non-convex and non-differentiable constraint, classical gradient-based algorithms like PGD (Madry et al., 2018) cannot be used for computing $\ell_0$-norm attacks. We categorize the existing $\ell_0$-norm attacks into two main groups: (i) multiple-norm attacks extended to $\ell_0$, and (ii) attacks specifically designed to optimize $\ell_0$ perturbations. Furthermore, we discuss related work that leverages the approximation of $\ell_0$ for different goals.

**Multiple-norm attacks extended to $\ell_0$.** These attacks are developed to work with multiple $\ell_p$ norms and include the extension of their algorithms to the $\ell_0$ norm. While they are able to find sparse perturbations, they often require strong use of heuristics to work in this setting. Brendel et al. (2019) initializes the attack from an adversarial example far away from the clean sample and optimizes the perturbation by walking with small steps on the decision boundary trying to get closest to the original sample. In general, the algorithm can be used for any $\ell_p$ norm, including $\ell_0$, but the individual optimization steps are very costly. Pintor et al. (2021) propose the Fast Minimum-Norm (FMN) attack that does not require an initialization step and converges efficiently with lightweight gradient-descent steps. However, their approach was developed to generalize over $\ell_p$ norms, but it does not make special adaptations to specifically minimize the $\ell_0$ norm. Matyasko & Chau (2021) use a two-player approach that optimizes the trade-off between perturbation size and loss of the attack and uses relaxations of the $\ell_0$ norm (e.g., $\ell_{1/2}$) to promote sparsity. This scheme however does not strictly minimize the $\ell_0$ norm, as the relaxation does not set the lowest components exactly to zero.

$\ell_0$**-specific attacks.** Croce et al. (2022) introduced SparseRS, a random search-based adversarial attack that explores potential perturbation candidates to return the highest confidence solution. Unlike minimum-norm attacks, their approach is rooted in a maximum-confidence attack framework with a predefined number of feature manipulations. Césaire et al. (2021) have designed an attack specifically for the $\ell_0$ norm. This attack is modeled as a stochastic Markov problem. It induces folded Gaussian noise to selected input components, iteratively finding the set that achieves misclassification with minimal perturbation. However, their approach requires a considerable amount of memory to explore the possible combinations and to find an optimal solution. This makes it infeasible to use for larger problems. With $\sigma$-zero, we show that the benefits from both groups, efficiency and precision, can be combined to effectively generate sparse $\ell_0$ attacks. It stands therefore as a promising solution for evaluating DNNs' robustness within the $\ell_0$ threat model, which remains relatively underexplored in existing benchmarks (Croce et al., 2021).

**Approximation of the $\ell_0$ norm.** Given the nonconvex and discontinuous nature of the $\ell_0$ norm, the adoption of surrogate approximation functions has been extensively studied (Bach et al., 2012; Weston et al., 2003; Zhang, 2008). Chen et al. (2018) use elastic-net regularization to calculate sparse perturbations, however, their attack do not necessarily find minimum $\ell_0$-norm perturbations. In our work, we use the formulation proposed by Osborne et al. (2000a), which provides an unbiased estimate of the actual $\ell_0$. Furthermore, it has been employed by Cinà et al. (2022) in the context of poisoning attacks to decrease sparsity in the model's activations, while we use it as a penalty term for crafting minimum $\ell_0$-norm adversarial examples.

## 5 CONCLUSIONS, LIMITATIONS, AND FUTURE WORK

Despite numerous proposed attacks for assessing DNN robustness, evaluation methods tend to overlook the significance of $\ell_0$-norm attacks (Chen et al., 2018; Croce & Hein, 2021a). However, these attacks can provide valuable insights into identifying the minimum manipulated input values required for successful attacks and reveal crucial information about model limitations. We argue that this literature gap is primarily due to the non-differentiable nature of the $\ell_0$ norm and its computational complexity, which poses challenges for gradient-based optimization.

In this work, we present $\sigma$-zero, a novel approach that leverages a smooth approximation of the $\ell_0$ norm. By making the objective differentiable, our method becomes amenable to optimization with gradient descent. Through extensive experimentation, we demonstrate the efficacy, precision, and scalability of $\sigma$-zero in diverse scenarios, specifically for identifying minimal $\ell_0$ perturbations. Our approach consistently discovers smaller minimum-norm perturbations across all models and datasets, while maintaining computational efficiency in execution time and VRAM consumption, and without requiring any computationally-demanding hyperparameter tuning. By identifying the smallest number of input values that can be modified to mislead the target model, our attack provides valuable insights on the vulnerabilities of DNN models and what they learn as salient input characteristics. Additionally, it may also provide meaningful insights on how to mitigate such vulnerabilities to improve robustness.

Although our approach offers promising results for benchmarking DNNs robustness, it relies on the white-box assumption. However, in the absence of such access, attackers may resort to techniques like transferability or gradient estimation to exploit vulnerabilities (Carlini et al., 2019; Tramèr et al., 2020). We acknowledge the significance of this analysis and plan to investigate it further in future research endeavors.

In conclusion, $\sigma$-zero emerges as a highly promising candidate for establishing a standardized benchmark to evaluate robustness against sparse $\ell_0$ perturbations. By facilitating more reliable and scalable assessments, it is poised to drive significant advancements in the development of novel models with improved robustness guarantees against the specific threat model under consideration.

**Ethics Statement.** Based on our comprehensive analysis, we assert that there are no identifiable ethical considerations or foreseeable negative societal consequences that warrant specific attention within the confines of this study. Rather this study will help improve the understanding of adversarial robustness properties of DNNs, and identify potential ways in which robustness can be improved.

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
