# OpenReview forum: "$\sigma$-zero: Gradient-based Optimization of \\$\ell_0$-norm Adversarial Examples"
_ICLR.cc/2024/Conference — Submitted to ICLR 2024_

### Official Review · Reviewer_sy67 · 2023-11-01

**Soundness:** 3 good
**Presentation:** 3 good
**Contribution:** 3 good
**Rating:** 6
**Confidence:** 4

**Summary:**

This paper propose a $l_0$-norm attack, called $\sigma$-zero, which leverages an differentiable approximation of the $l_0$  norm to facilitate gradient-based optimization.

**Strengths:**

This paper's primary contribution lies in its application of the $l_0$ norm approximation function, as introduced by Osborne et al. (2000b), to $l_0$ attacks. The research presents thorough experiments on various robust models (e.g., C1-C8) presented in Robustbench and multiple datasets.

When compared with existing sparse attacks, the results convincingly demonstrate that sigma-zero outperforms in terms of attack rates and offers reduced computational costs.

While the authors have included the code in the supplementary materials, I haven't personally tested it. Nonetheless, I anticipate that the broader community will benefit once the authors make their code publicly available in the future.

**Weaknesses:**

A primary shortcoming of the paper is its resemblance to an experimental or technical report rather than a comprehensive academic study.

The discussion about the scientific principles about why $\sigma$-zero performs better is quite sparse in the current presentation.

Additionally, the term "VRAM" is not defined. It would enhance clarity if its full name were provided initially for the readers not familiar with.

**Questions:**

see weakness.

---

> ### Author Response · Authors · 2023-11-15
> **Response to Reviewer sy67**
>
> We thank the reviewer for acknowledging the value of our contribution and its benefit to the community.
>
> **Writing style**. We acknowledge the constructive criticism of the reviewer, and will improve our writing style by better highlighting the contributions and novelty of our work, as well as the reasons behind why the attack works well. Our purpose was genuinely to make our work reproducible, as also encouraged by the conference policy. This is why we reported too many results and technical details in the paper. However, we are committed to enhancing the paper's accessibility and will relocate the more technical details and results to the appendix.
>
> **sigma-zero results.** We will improve the paper emphasizing the main contributions and condensing their descriptions in a well-identifiable paragraph in the paper. In particular, we posit that several factors contribute to the effectiveness of $\sigma$-zero, including the type of approximation used for the L0 norm, the novel loss formulation presented in Eq. 7, and the incorporation of gradient normalization in our optimization process. To elaborate further, we will clarify how the L0-norm approximation can enhance the effectiveness of our approach, offering a numerically stable method for minimizing L0 in adversarial perturbations. Additionally, we will detail the characteristics of the loss formulation introduced in Equation 7, explaining that it is carefully designed such that its solution corresponds to a minimum L0-norm perturbation. Finally, we will delve into the role of gradient normalization towards ensuring convergence during our optimization process.
> &nbsp;
>
> **VRAM.** We will also revise the paper to make clear that VRAM refers to Video Random Access Memory, and its role. Specifically, VRAM is a type of memory designed explicitly for use in graphics processing units (GPUs). In machine learning, GPUs have become increasingly important due to their parallel processing capabilities, making them well-suited for tasks involving large-scale mathematical computations, such as those found in machine learning algorithms.
>
> We will add these considerations to the paper and would be happy to know if the reviewer has any other requests. We would appreciate a score increase from the reviewer if he/she finds our clarifications satisfactory. We remain available for further clarification.

---

> > ### Comment · Reviewer_sy67 · 2023-11-23
> > **Thanks for the response**
> >
> > Thanks for the response.
> >
> > I gone through the opinion from other reviewers and realize the issue raised in the experiments on Sparse RS. I appreciate the effort made by the authors in addressing this issue. I intend to keep my score.

---

> > > ### Author Response · Authors · 2023-11-23
> > > **Thank you**
> > >
> > > Thanks for your kind response, and all the efforts made in reviewing our work even in its post-rebuttal version. We appreciated that. Let us remark that we have improved significantly the performance of $\sigma$-zero which turned out to be even a more effective attack. We have hopefully finally addressed all the issues raised by the reviewers, including all comments on Sparse-RS, additional model-attack evaluations, etc. It has been a huge effort to address all these comments in the last weeks, and we hope that the reviewers will favorably recognize not only our effort, but also the merit of the proposed approach. Indeed, we firmly believe that $\sigma$-zero provides an additional, powerful tool to improve further the reliability of empirical adversarial robustness evaluations against sparse, L0-norm attacks, which may have future important implications in explainability analyses, to gain a better understanding about what DNN models learn as salient features to discriminate samples from different classes.

---

### Official Review · Reviewer_oGkp · 2023-11-02

**Soundness:** 3 good
**Presentation:** 3 good
**Contribution:** 3 good
**Rating:** 5
**Confidence:** 5

**Summary:**

This paper proposes $\sigma$-zero attack, a sparse attack gained by minimizing the norm-0 of the perturbation.

**Strengths:**

- The idea is simple and intuitive.

- The experimental results look good.

**Weaknesses:**

-  The adversarial images created by the proposed approach do not look visually appealing according to Figure 3. It is easy to spot out highly-intensive pixels.

-  The purpose of adversarial attacks is to generate visually appealing images that is imperceptible to human vision. Therefore, less norm-0 perturbations do not mean better visually appealing adversarial images. I feel that although this approach can help to restrict the number of pixels perturbed, it tends to perturb other pixels more, leading to adversarial images as in Figure 3.

- The norm-0 solely is not adequate to measure the quality of generated adversarial images. The norm-0 of the proposed approach is smaller because it directly minimize an approximation of the norm-0. It would be better if the paper reports some other metrics such as  SSIM, PSNR, and LPIPS.

- Moreover, the robust classifiers in RobustBench are trained mostly with $\ell_\infty$ and $ell_2$, hence they cannot defend well the proposed approach. What does it happen if we train a robust classifier with 20-steps $\sigma$-zero and then evaluate on the same attack with 100 steps?

**Questions:**

Please refer to the weakness section.

---

> ### Author Response · Authors · 2023-11-15
> **Response to Reviewer oGkp**
>
> We thank the reviewer for acknowledging the simplicity and effectiveness of our proposed attack, and reply to their concerns below.
> &nbsp;
>
> **Visually-imperceptible Adversarial Examples.** We aim to clarify with the reviewer that visual imperceptibility of adversarial examples is considered now a known misconception [S0,S1]. In fact, as highlighted by the examples reported in [S0,S1], including image spam and visibly-modified traffic signs, the perturbations are clearly visible to the human eye (even though the main content is still preserved), as their sole purpose is to evade automatic detection. In addition, quoting Justin Gilmer et al. [S0]: “at the time of writing, we were unable to find a compelling example that required indistinguishability.” Recall indeed that indistinguishability is a much stricter requirement than just preserving the main content (e.g., the ability of humans to still recognize the correct object in the image). This is also consistent with the definition of adversarial example given by Ian Goodfellow, which does not even mention the requirement of visual indistinguishability: “Adversarial examples are inputs to machine learning models that an attacker has intentionally designed to cause the model to make a mistake”. Accordingly, we can say that the goal of an adversarial example is simply to fool the machine-learning model, and not necessarily be imperceptible to the human eye. This is especially true for sparse (L0 and L1) attacks, as they are constrained to only manipulate a few input values, but it is also an issue that affects all previous work related to sparse attacks, not just ours [S2, S3, S4, S5]. In fact, by inspecting the attacks proposed in previous work, it is not difficult to see that L0 and L1 adversarial perturbations are always clearly visually distinguishable (see, e.g., Figure 3 in our paper). Their goal, indeed, is not to be indistinguishable to the human eye, but rather show whether and to which extent models can be fooled by just changing a few input values -- i.e., to evaluate their adversarial robustness against sparse perturbations. We hope that the reviewer agrees with this perspective and, if necessary, we will take this opportunity to clarify this aspect in the paper too.
>
> [S0] J. Gilmer, R. P. Adams, I. J. Goodfellow, D. Andersen, and G. E. Dahl. Motivating the rules of the game for adversarial example research. CoRR, abs/1807.06732, 2018.
>
> [S1] Biggio, Battista and Fabio Roli. “Wild Patterns: Ten Years After the Rise of Adversarial Machine Learning.” Proceedings of the 2018 ACM SIGSAC Conference on Computer and Communications Security (2017).
>
> [S2] Croce, Francesco and Matthias Hein. “Mind the box: l1-APGD for sparse adversarial attacks on image classifiers.” International Conference on Machine Learning.
>
> [S3] Modas, Apostolos et al. “SparseFool: A Few Pixels Make a Big Difference.” 2019 IEEE/CVF Conference on Computer Vision and Pattern Recognition (CVPR).
>
> [S4] Croce, Francesco et al. “Sparse-RS: a versatile framework for query-efficient sparse black-box adversarial attacks.” AAAI Conference on Artificial Intelligence (2020).
>
> [S5] Su, Jiawei et al. “One Pixel Attack for Fooling Deep Neural Networks.” IEEE Transactions on Evolutionary Computation 23 (2017).
>
> &nbsp;
>
> **Robust Models.** As suggested by reviewer g7Nu, we now have integrated two more models trained to be robust against sparse attacks [S6, S7]. The results have been reported above in  our response to all reviewers (https://openreview.net/forum?id=dXRWP4n15q&noteId=JKLyN3Qnl9).
> Even against such robust models, $\sigma$-zero continues to outperform existing attacks, exhibiting higher attack success rates, lower distances, and faster execution. We will add these experiments and considerations to the paper and would be happy to know if the reviewer has any other requests.
>
> [S6] Croce, Francesco, and Matthias Hein. "Mind the box: -APGD for sparse adversarial attacks on image classifiers." International Conference on Machine Learning. PMLR, 2021.
>
> [S7] Jiang, Yulun, et al. "Towards Stable and Efficient Adversarial Training against  Bounded Adversarial Attacks." International Conference on Machine Learning. PMLR, 2023.
> &nbsp;
>
> We would appreciate a score increase from the reviewer if he/she finds our clarifications satisfactory. We remain available for further clarification.

---

> ### Comment · Reviewer_oGkp · 2023-11-23
>
> Thanks the authors for your feedback. I do not agree with your point of the unnecessity of maintaining visual imperceptibility of adversarial examples. If it happens, why you need to minimize the $\ell_0$ of the perturbation. If we relieve such constraints, we can easily find the adversarial examples with attack successful rate 100%. It would be more convincing to me if the authors can give a practical scenario of using the easily human-recognized adversarial examples. Moreover, it would be better if the paper reports some other metrics such as SSIM, PSNR, and LPIPS. I am leaning to keep my current score.

---

> > ### Author Response · Authors · 2023-11-23
> > **Response to Reviewer oGkp**
> >
> > We thank the reviewer for the respose.
> >
> > We aim to minimize the perturbation to understand how many features must be altered to evade the target model. However, the perturbation does not necessarily have to be imperceptible. Otherwise, the existing previously proposed patch attacks [A1, A2], physical attacks [A3], and universal perturbation attacks [A4, A5] would not make any sense. Only earlier attacks using $\ell_2$ and $\ell_\infty$ in the digital domain were claimed to preserve imperceptibility, but only against undefended models. So, we do believe that studying other attacks is also widely relevant.
> >
> > Besides, we can still evaluate SSIM and other metrics and add them to the paper, along with a discussion about the imperceptibility of $\ell_0$ attacks if this would fulfill the reviewer's request.
> >
> >
> > [A1] Karmon et al. Localized and visible adversarial noise" International Conference on Machine Learning (2018).
> >
> > [A2] Brown et al. "Adversarial patch" ArXiv (2017).
> >
> > [A3] Eykholt et al. "Robust Physical-World Attacks on Deep Learning Visual Classification." IEEE/CVF Conference on Computer Vision and Pattern Recognition (2018).
> >
> > [A4] Moosavi-Dezfooli et al. "Universal Adversarial Perturbations." IEEE Conference on Computer Vision and Pattern Recognition (2016).
> >
> > [A5] Benz et al. "Double targeted universal adversarial perturbations" Asian Conference on Computer Vision (2020).

---

> ### Author Response · Authors · 2023-11-23
> **SSIM and PSNR evaluation**
>
> We have been able to run some more additional experiments, reported in the supplementary material (see comparison_SSIM_PSNR.pdf),  where we computed SSIM and PSNR measures for different examples. Note that only EAD exhibits marginally superior values w.r.t. the other attacks, given that it works using a slightly different perturbation model.
>
> The proposed $\sigma$-zero  attack instead achieves very high SSIM and PSNR, outperforming or remaining comparable to the other $\ell_0$-norm attacks.
>
> Let us finally thank again the reviewer for giving us the opportunity to improve our work. We hope that this last evaluation will convince the reviewer that our attack is valuable and that it can improve our current understanding of adversarial robustness against sparse, $\ell_0$-norm attacks.

---

### Official Review · Reviewer_nLC7 · 2023-11-02

**Soundness:** 3 good
**Presentation:** 3 good
**Contribution:** 3 good
**Rating:** 5
**Confidence:** 5

**Summary:**

The paper proposes $\sigma$-zero, a white-box adversarial attacks for the $\ell_0$-threat model. In particular, $\sigma$-zero obtains sparse perturbations minimizing a differentiable surrogate of the $\ell_0$-norm. In the experiments on several datasets and target classifiers, $\sigma$-zero is shown to outperform existing attacks in terms of success rate and and size of the perturbations.

**Strengths:**

- The proposed approach of using a differentiable approximation of the $\ell_0$-norm is reasonable, and yields a simple method.

- The effectiveness of $\sigma$-zero is supported by the experimental results.

**Weaknesses:**

- The configuration in which some of the competitors are used seems suboptimal:
    - BB can be initialized from an image of another class or another dataset, as done in [A, B], to avoid the issue of not finding a starting point (as mentioned in Sec. 3.2).
    - If I understand it correctly, Sparse-RS is re-run with different sparsity levels until reaching the desired success rate (averaged over the test points), but then the results are reported for all points with the same sparsity level. In this case, the results would be suboptimal, as the binary search should ideally be done for each point individually (for comparison to $\sigma$-zero and other attacks which optimize the perturbation size independently for each test image). As a cheaper solution, one could run Sparse-RS on several $k$ values, and the select for each point the smallest $k$ which finds an adversarial perturbations.
    - In general, while the paper uses the default parameters for all baseline attacks, it's not clear whether these are optimal: ideally, one could tune (some of) them on a small subset of test cases (models or datasets).
    - Sparse-RS is a black-box method, i.e. doesn't need a backward pass at each iteration, which means that using the same number of iteration as for the white-box attacks results in significantly lower computational cost (e.g. 2x fewer network passes). How would the results compare when equating the number of network passes?

- The overall technical contribution is limited: while the proposed algorithm has some task-specific solutions, e.g. inducing sparsity in $\delta$ by clipping the smallest components, the $\ell_0$-norm approximation has already been used in the context of adversarial robustness (Cinà et al., 2022).

[A] https://arxiv.org/abs/2102.12827
[B] https://arxiv.org/abs/2103.01208

**Questions:**

Since the experimental results are the main part of the paper, I think it is important that the configuration in which the baselines are used is clarified to provide a comprehensive comparison and assess the effectiveness of the proposed method.

---

> ### Author Response · Authors · 2023-11-15
> **Response to Reviewer nLC7**
>
> We thank the reviewer for acknowledging the significant results sigma-zero offers over all the experimental settings we provide.
>
> **Additional comparison with BB.** We have run additional experiments considering BB with adversarial initialization (BBadv), whose results are reported in  our response to all reviewers (https://openreview.net/forum?id=dXRWP4n15q&noteId=JKLyN3Qnl9). These experiments further confirm the effectiveness of sigma-zero against competing attacks, in terms of ASR, perturbation size, and execution time. We also remark that, over five additional experimental configurations, only once BBadv finds slightly better results (i.e., 3 features less in median perturbation size). However, this improvement comes at the cost of a 12-fold increase in execution time, raising the computational complexity from 0.42 seconds per sample to 5.28 seconds. Furthermore, BBadv exploits an unfair initialization strategy compared to the other attacks, given that all the other ones start from the same source sample, and not from an adversarial region. We will add these experiments and considerations to the paper.
>
> **Comparison with Sparse-RS.** We agree with the reviewer that SparseRS was tested in an unfair setting. We have thus re-run it using a sample-wise binary search on the perturbation budget, to efficiently find minimum-norm adversarial examples, as suggested in [S0], and as also detailed in the response to reviewer g7Nu (see **Unfair comparison with Sparse-RS**, https://openreview.net/forum?id=dXRWP4n15q&noteId=9oqEGzhkh4 ). Even under this configuration, $\sigma$-zero shows better results and computational efficiency. We’d finally like to remark also that SparseRS should not be seen as a direct competitor of gradient-based (white-box) attacks, including $\sigma$-zero, given its gradient-free (black-box) nature. We report it in our experiments as an additional, interesting baseline, and will clarify this aspect in the paper too. Note finally that the number of network passes **(i.e. forwards and backwards)** is already set to be the same between white-box and black-box attacks. This means that SparseRS is already performing twice the number of forward passes w.r.t. gradient-based attacks. We'll clarify this aspect in the paper.
>
> [S0] Rony, Jérôme et al. “Augmented Lagrangian Adversarial Attacks.” 2021 IEEE/CVF International Conference on Computer Vision (ICCV).
>
> **Hyperparameter optimization.** We agree in principle with the reviewer that optimizing the attack hyperparameters may be beneficial for some attacks. However, this is normally too costly and not considered in the majority of papers dealing with attack and defense evaluations (see, e.g.,  [S0, S1, S2, S3, S4, S5]). This is also why AutoPGD (and AutoAttack) algorithms are so popular, namely, they are parameter-free and work well with their default hyperparameter values. For this reason, we also consider here the same setting of using the default hyperparameter values reported in the original papers/implementations of each attack, and show that without any specific hyperparameter tuning, $\sigma$-zero turns out to be more effective. In this sense, we do believe that our comparison remains fair (namely, all attacks use default hyperparameters, and no computational overhead is spent in hyperparameter tuning).
>
> [S1] Croce, Francesco and Matthias Hein. “Mind the box: l1-APGD for sparse adversarial attacks on image classifiers.” ICML.
>
> [S2] Hajri, Hatem et al. “Stochastic sparse adversarial attacks. 2021 IEEE 33rd ICTAI.
>
> [S3] Rony, Jérôme et al. “Decoupling Direction and Norm for Efficient Gradient-Based L2 Adversarial Attacks and Defenses.” 2019 IEEE CVPR.
>
> [S4] Modas, Apostolos et al. “SparseFool: A Few Pixels Make a Big Difference.” 2019 IEEE/CVF CVPR.
>
> [S5] Croce, Francesco and Matthias Hein. “Sparse and Imperceivable Adversarial Attacks.” 2019 IEEE/CVF ICCV.
>
>
> **Paper’s contributions.** Finally, we apologize to the reviewer for not having highlighted our contributions well within the paper. Specifically, the loss term in Eq. 7 represents a novel contribution as it enables us to simultaneously search for an adversarial example while also minimizing the L0 norm of the perturbation (i.e., a non-trivial task given the non-convexity of this norm). Despite its simplicity, the approximation function used in Cinà et al. enables the implementation of a simple yet very fast and surprisingly effective attack, as witnessed by our extensive experimental analysis and acknowledged by the other reviewers. We thus firmly believe that the idea of leveraging such an approximation to develop a very effective L0-norm adversarial attack is not trivial.
>
> We would appreciate a score increase from the reviewer if he/she finds our clarifications satisfactory. We remain available for further clarification.

---

> > ### Comment · Reviewer_nLC7 · 2023-11-21
> >
> > I thank the authors for the response and additional experiments.
> >
> > When using initialization BB performs quite close (sometimes better, interestingly on the most robust model) to $\sigma$-zero, although I agree its cost is higher. I don't think the initialization gives a particular advantage to BB, since I assume it's completely non-sparse (this could be verified just starting the other attacks from the same point).
> >
> > Also, since in the new results SparseFool gets median $||\delta||_0$ up to 3071, it means that the considered threat model counts color channels independently, instead of counting just number of perturbed pixels. I think it's worth clarifying this aspect in the paper (unless it's already there and I missed it) since there are some attacks use the pixel space (or e.g. Sparse-RS handles both, and I assume it's used in the feature space mode -- is that correct?).
> >
> > Overall, I'm leaning towards keeping the original score.

---

> > > ### Author Response · Authors · 2023-11-21
> > > **Response to Reviewer nLC7**
> > >
> > > Dear reviewer,
> > >
> > > first of all, thanks for your reply and additional comments. We'd like to point out that we have just uploaded a deeply revised version of our paper as the main file, and a diff.pdf file highlighting all the changes made in the supplementary material.
> > >
> > >
> > > We have improved our attack by dynamically adjusting the sparsity threshold (per sample), and re-run all the experiments.
> > > Our novel results demonstrate a significant improvement over the earlier version of $\sigma$-zero, which now even outperforms both BB and BB with adversarial initialization (BBadv) on all cases, finding adversarial examples with much smaller perturbations and, most importantly, much faster.
> > >
> > > We have also clarified that all attacks, including Sparse-RS, are configured to modify all input values (not restricted to individual pixel changes), and confirm that Sparse-RS is also configured to perform twice the number of iterations.
> > >
> > > We hope that, in light of the last improvements to our paper, you can reconsider your evaluation or help us understand what can still be done to improve our work. The experiments in our paper already show the superiority of $\sigma$-zero against competing approaches, in a very wide set of diverse configurations, making it the de-facto state of the art for $\ell_0$ attacks.
> > >
> > > Let us conclude by thanking you again for your dedication and the effort spent reviewing our work.

---

> > > > ### Comment · Reviewer_nLC7 · 2023-11-23
> > > >
> > > > Thanks for the update. The revised version differs quite significantly from the original, i.e. all major parts of the paper (proposed method, baselines, main experiments) have been modified. This has some side effects: for example, the impact of the new thresholding scheme is not well discussed, and I guess the ablation study in the appendix is now outdated. Moreover, Table 1 has a fair amount of missing results, and the comparison on ImageNet misses some of the strongest baselines.
> > > >
> > > > As additional question, when using binary search for maximum confidence attacks, is the iteration budget split across the different perturbation sizes used or kept the same (i.e. 1000 or 2000 iterations for each $k$)?

---

> ### Author Response · Authors · 2023-11-23
> **Response to Reviewer nLC7**
>
> We appreciate the reviewer for their response.
>
> **Thresholding scheme**. The main modification of the thresholding scheme aims to provide a sample-wise threshold tuning which drastically improved the performance with respect to having a fixed threshold equal for all samples.
>
> **Ablation study.** The reported ablation study, as detailed in the appendix (refer to Figure 5), is not outdated. It has been updated to incorporate the updated version of our attack including the new thresholding scheme. We demonstrate that the initial value selection for the thresholding parameter, denoted as $\tau$, has negligible influence on the outcome, given that the parameter dynamically adapts throughout the optimization process.
>
> **Binary Search.** Regarding binary search, yes, the iteration budget is split across different perturbation sizes.

---

> > ### Comment · Reviewer_nLC7 · 2023-11-23
> >
> > > Binary Search. Regarding binary search, yes, the iteration budget is split across different perturbation sizes.
> >
> > How does this work in practice then? I think this should be described somewhere.

---

> ### Author Response · Authors · 2023-11-23
> **Response to Reviewer nLC7**
>
> We'll detail that aspect better in the paper, but we essentially followed the same mechanisms explained in [S0].
> We imposed 400 and 100 number of steps respectively for Sparse-RS and PGD-$\ell_0$.
>
> As depicted in Table 1 (column q), with these configurations, the two attacks execute approximately 2000 queries or more each. Consequently, they exploit, on average, the same number of queries compared to the other attacks, ensuring a fair comparison.
>
> [S0] Rony, Jérôme et al. “Augmented Lagrangian Adversarial Attacks.” 2021 IEEE/CVF International Conference on Computer Vision (ICCV).

---

> > ### Comment · Reviewer_nLC7 · 2023-11-23
> >
> > Thanks for the clarification. I think this effectively limits the performance of the baselines: in fact only a small number of iterations are used for each $k$, and only a limited number of $k$ values can be tested (because of the overall budget of iterations), which might make the initial value of $k$ quite influential on the final results.

---

> ### Author Response · Authors · 2023-11-23
> **Response to Reviewer nLC7**
>
> Recall that PGD-L0 has 100 iterations by default, and that increasing the iterations of PGD-L0 and Sparse-RS for each k would then be unfair to minimum-norm attacks. However, if this is important, we can also include results with more iterations for these attacks too, or even better, compare ASR at fixed k, using the same number of iterations for sigma-zero, Sparse-RS, and PGD-L0. Despite this scenario being slightly unfair to sigma-zero, we are confident that sigma-zero will outperform the other attacks also in that configuration. Finally, we express again our sincere gratitude for your insights and feedback. We hope that we have provided convincing evidence that our attack works very well, especially post rebuttal, and that it can thus provide an additional, sound algorithm to help improve adversarial robustness evaluations in the L0 case.

---

### Official Review · Reviewer_g7Nu · 2023-11-05

**Soundness:** 3 good
**Presentation:** 2 fair
**Contribution:** 2 fair
**Rating:** 6
**Confidence:** 5

**Summary:**

This paper proposes an algorithm utilizing gradient information to generate sparse adversarial perturbations. Compared with perturbations bounded by $l_2$ or $l_\infty$ norms, sparse perturbation is more challenging because of its non-convexity nature. The authors use a continuous function to approximate the $l_0$ norm of the perturbation and design a new loss objective function that facilitates optimizing adversarial perturbations. The experiments show the effectiveness and efficiency of the proposal algorithm.

**Strengths:**

Robustness against sparse perturbation is an interesting problem to explore. The algorithm is well-motivated and clearly demonstrated. The experiments are conducted on various datasets and the results indicate the advantages of the proposal algorithm over the baselines considered.

**Weaknesses:**

1. Although the experiments are conducted in various datasets, I think more sparse attack algorithms should be included as the baselines for comparison. For example, PGD$_0$ [A] should be included as the baseline, since it is also a white-box attack for $l_0$ bounded perturbations. Sparsefool, based on constructing sparse perturbations on top of popular deep fool method, should also be studied.

[A] Francesco Croce and Matthias Hein. Sparse and imperceivable adversarial attacks. In Proceedings of the IEEE/CVF international conference on computer vision, pp. 4724–4732. 2019.

[B] Modas, Apostolos, Seyed-Mohsen Moosavi-Dezfooli, and Pascal Frossard. "Sparsefool: a few pixels make a big difference." Proceedings of the IEEE/CVF conference on computer vision and pattern recognition. 2019.

2. The comparison might not be fair. Some algorithms are proposed in a different formulation as in this paper. Some baselines (such as Sparse-RS) are proposed to generate adversarial examples such that the $l_0$ norms of the perturbations are smaller than $\epsilon$. These algorithms are not designed to minimize the $l_0$ of the perturbation, they search for a perturbation whose $l_0$ norm is smaller than a threshold. Comparing the $l_0$ norm - ASR trade-off is probably unfair for these methods.

3. In addition to $l_2$ and $l_\infty$ robust models, I think including $l_1$ robust model for evaluating the attacks would make the experiment more comprehensive, especially considering the $l_1$ norm is the closest convex $l_p$ norm to $l_0$ norm. Possible baselines include those trained by AA-$l_1$ [C] and Fast-EG-$l_1$. [D]

[C]: Croce, Francesco, and Matthias Hein. "Mind the box: $ l_1 $-APGD for sparse adversarial attacks on image classifiers." International Conference on Machine Learning. PMLR, 2021.

[D]: Jiang, Yulun, et al. "Towards Stable and Efficient Adversarial Training against $ l_1 $ Bounded Adversarial Attacks." International Conference on Machine Learning. PMLR, 2023.

**Questions:**

Major concerns are demonstrated in the weakness part. In addition to these concerns, I have the following questions:

1. Function $\mathcal{L}$, as defined by Equation (7), is not continuous and thus not differentiable everywhere. Will this cause some problems when calculating the gradient of $\mathcal{L}$ in line 4 of Algorithm 1? I think using a continuous and differentiable function as the loss objective would be better.

2. Regarding Sparse-RS with a super-script 100 or 85. If understood correctly, these super-scripts mean the value of k, max allowed $l_0$ norm of the perturbations, why Sparse-RS100 is worse than Sparse-RS85? And why the average $|\delta|_0$ is bigger than the corresponding k in some cases?

Due to the major concerns and questions as pointed out, I cannot recommend acceptance. I welcome the discussions with the authors and will reconsider my rating.

---

> ### Author Response · Authors · 2023-11-15
> **Response to Reviewer g7Nu**
>
> We thank the reviewer for acknowledging credit for our contribution and its clarity, and reply to their concerns below.
>
> **Additional attacks and robust models.** We have run additional experiments considering the additional attacks PGD-L0 [A] and Sparsefool [B], and the L1-norm robust models [C] and [D], as suggested by the reviewer. We report the results in our response to all reviewers (https://openreview.net/forum?id=dXRWP4n15q&noteId=JKLyN3Qnl9), showing that $\sigma$-zero outperforms such attacks on the suggested robust models as well as on the other CIFAR10 models from our paper. We will add these results to the revised paper, as well as the comparisons on the remaining models we used in the paper (which will take some more time to execute). With these additional results, our work will include more than 320 diverse dataset-model-attack configurations, which we believe should provide sufficient convincing empirical evidence that our attack achieves state-of-the-art performance.
>
> **Unfair comparison with Sparse-RS**. We agree with the reviewer that comparing minimum-norm attacks against maximum-confidence attacks (i.e., attacks that optimize the loss within a maximum perturbation budget, like PGD-L0 and Sparse-RS) may be tricky. In particular, minimum-norm attacks are typically more complicated as they require not only finding an adversarial example, but also the minimum perturbation budget required to do that (i.e., they optimize both the loss and the perturbation size, not only the loss).
> To compare these two types of attack in a fair manner, we have re-run our experiments by performing a binary search on the perturbation budget of **each sample** when using maximum-confidence attacks (rather than doing that per batch), as also suggested in [S1]. In addition, we do not only report the median distance (i.e., the distance that corresponds to an ASR of 50%), but also the ASR at a fixed budget with eps=10 and eps=50, as typically done when evaluating maximum-confidence attacks. Given that we re-evaluated Sparse-RS within this setting, we decided to remove the two redundant SparseRS versions achieving different ASRs (85% and 100%, respectively).
>
> **Loss gradient.** Regarding Eq 7, we agree with the reviewer that using a sigmoid-like function instead of a step function may be more appropriate. We have however seen in practice that this never induces numerical instabilities on the gradient even on the 18 models we consider in our experimental analysis. We will clarify this point within the paper, after Eq. 7.
>
> **Superscripts in Sparse-RS.** First, let us point out that comparing against Sparse-RS is only meant to provide an additional baseline, but it should not be considered a direct competitor of gradient-based (white-box) attacks, given that Sparse-RS is a gradient-free (black-box) attack (working under stricter assumptions). As explained before, the superscripts in Sparse-RS were not referred to the perturbation budget, but to the ASR. This means that the perturbation budget of Sparse-RS^85 and Sparse-RS^100 was tuned to respectively achieve an ASR of 85% and 100%. However, given that we now tune the perturbation budget in a sample-wise manner to find minimum-norm adversarial examples also for SparseRS, we decided to remove these two (confusing) versions of SparseRS from our evaluation.
>
> We would appreciate a score increase from the reviewer if he/she finds our clarifications satisfactory. We remain available for further clarification.

---

> > ### Comment · Reviewer_g7Nu · 2023-11-23
> > **Response**
> >
> > I thank the author for comprehensive feedback. I agree with the author that minimum-norm attacks are typically more challenging than the attack with a fixed norm requirement. However, as far as what I know, there is no algorithm currently to **guarantee** the minimum $l_0$ norm is achieved to generate adversarial examples. By contrast, there are indeed several methods that can *successfully* generate adversarial examples bounded by a specific level of $l_0$ norm.
> >
> > I encourage the author to clarify the difference between these two categories of methods to avoid confusion.
> >
> > For the rest questions, it is well addressed. The score is adjusted accordingly.

---

### Author Response · Authors · 2023-11-15
**Response to all reviewers: Additional experiments**

We thank the reviewers for acknowledging credit for our contribution and for providing us constructive feedback.

We here report the results of the additional experimental comparison suggested by the reviewers, including additional attacks (i.e., PGD-L0, BB with adversarial initialisation, and sparsefool) and models robust to sparse perturbations (i.e., [C] and [D]). In particular, we compare the additional attacks with the most representative baselines on the suggested models and on the CIFAR10 models [C2], [C3], and [C4]. We are running the remaining experiments and will include results for all models in the paper.
Recall that $\sigma$-zero continues to outperform most of the competing  attacks even on the models robust to sparse perturbations.

|Model                    |Attack Name|ASR (%)|ASR (%)$_{10}$|ASR (%)$_{50}$|$&#124;&#124;\delta&#124;&#124;_0$|q (x1000)|t(s) |VRAM|
|-------------------------|-----------|-------|--------------|--------------|--------------------|----------------|-----|----|
|Croce et al. [C]  AA-l1  |FMN        |100    |5.66          |35.92         |59                  |2.0             |0.31 |0.59|
|                         |SPARSE-RS  |100    |4.93          |21.61         |99                  |2.41            |2.13 |0.69|
|                         |PGD-L0       |100    |4.10          |23.91         |78                  |1.91            |1.90 |0.72|
|                         |SPARSEFOOL |58.29  |2.94          |11.42         |3069                |0.47            |2.65 |0.65|
|                         |BBadv      |100    |6.88          |59.48         |38                  |2.01            |5.28 |0.65|
|                         |**$\sigma$-zero**|100    |6.47          |54.91         |41                  |2.0             |0.42 |0.65|
|Jiang et al. [D]  Fast-EG|FMN        |100    |8.85          |39.18         |49                  |2.0             |0.35 |0.59|
|                         |SPARSE-RS  |100    |6.83          |24.79         |81                  |2.03            |6.28 |0.69|
|                         |PGD-L0       |100    |4.42          |21.64         |84                  |1.85            |4.44 |0.71|
|                         |SPARSEFOOL |84.45  |3.06          |19.18         |92                  |0.47            |1.61 |0.66|
|                         |BBadv      |100    |11.89         |58.68         |31                  |2.01            |4.06 |0.65|
|                         |**$\sigma$-zero**|100    |11.76         |56.16         |31                  |2.0             |0.46 |0.65|
|C4                       |FMN        |100    |26.85         |85.6          |23                  |2.0             |1.09 |1.80|
|                         |SPARSE-RS  |100    |13.65         |53.01         |39                  |2.26            |13.2 |2.20|
|                         |PGD-L0       |100    |10.73         |60.87         |36                  |1.92            |8.97 |2.30|
|                         |SPARSEFOOL |82.71  |6.3           |29.59         |3036                |0.67            |3.74 |1.90|
|                         |BBadv      |100    |27.9          |89.94         |16                  |2.01            |4.51 |1.99|
|                         |**$\sigma$-zero**|100    |39.13         |98.53         |14                  |2.0             |1.44 |1.89|
|C3                       |FMN        |100    |20.61         |71.7          |33                  |2.0             |1.08 |1.80|
|                         |SPARSE-RS  |100    |12.49         |52.08         |39                  |2.26            |12.22|2.20|
|                         |PGD-L0       |100    |8.41          |49.9          |45                  |1.93            |10.30|2.30|
|                         |SPARSEFOOL |94.26  |5.34          |11.67         |3071                |0.26            |3.76 |1.90|
|                         |BBadv      |100    |22.72         |87.39         |18                  |2.0             |4.63 |1.99|
|                         |**$\sigma$-zero**|100    |32.1          |96.23         |16                  |2.0             |1.45 |1.89|
|C2                       |FMN        |100    |28.43         |74.7          |26                  |2.0             |0.59 |1.31|
|                         |SPARSE-RS  |100    |23.7          |59.5          |27                  |2.03            |6.60 |1.87|
|                         |PGD-L0       |100    |11.04         |46.68         |45                  |1.92            |5.8  |1.75|
|                         |SPARSEFOOL |47     |7.16          |7.44          |3070                |0.35            |4.03 |1.62|
|                         |BBadv      |100    |27.33         |80.26         |18                  |2.01            |4.98 |1.64|
|                         |**$\sigma$-zero**|100    |42.62         |94.62         |13                  |2.0             |0.86 |1.47|

---

### Author Response · Authors · 2023-11-21
**Request for feedback**

Dear AC, and respected reviewers, we kindly request you to share your thoughts about our paper. We've re-run all the experiments, and we are going to update the modified paper, including all the results that we've been able to compute so far. We've also improved our attack by adjusting the thresholding operation (line 7 in Algorithm 1) in a sample-wise manner, drastically improving the performance over all the datasets and models (including outperforming all attacks on the robust models suggested by the reviewers).

---

### Author Response · Authors · 2023-11-21
**Paper Revision and Experimental Updates**

We appreciate the reviewers' feedback, which has been fundamental in improving our paper. We've uploaded a revised version as the main file, along with supplementary materials containing the appendix, source code, and a "diff.pdf" file illustrating all changes made.

In summary, we have improved our attack by dynamically adjusting the sparsity threshold (per sample), and re-run all the experiments. Some experimental configurations, involving larger DDNs and slower attacks, are still running and will be available only next week.
Our novel results demonstrate a significant improvement over the earlier version of $\sigma$-zero, which now even outperforms both BB and BB with adversarial initialization (BBadv) on all cases, finding adversarial examples with much smaller perturbations and, most importantly, much faster.

We would appreciate a score increase from the reviewers if they find our clarifications and results satisfactory. We remain available for further clarification.

---

### Meta-Review · Area_Chair_8vZJ · 2023-12-10

**Metareview:**

The authors propose a minimum-norm $\ell_0$-attack. For this purpose they employ a loss function which employs a previously proposed $\ell_0$-approximation (which has also been used in a previous $\ell_0$-attack). During rebuttal the authors added a lot of comparisons to competing methods and tried to make the comparison more fair to attacks using an $\ell_0$-budget (but see comment below).

Strengths:
- the attack seems to work more efficiently than previous $\ell_0$-attacks (at least in the settings currently investigated and with the restriction discussed in weaknesses)

Weaknesses:
- the paper and the method has been  significantly changed during the rebuttal
- during the rebuttal it turned out that the comparison to budget based $\ell_0$-attacks is not fair as the total budget of iterations/queries seems to be divided over the different values of $k$ which is tried in the binary search.
- the results table is half-empty for Cifar10 and there is no comparison to strong competitors like Sparse-RS and BB on ImageNet
- reviewers complained that several details of the attack are unclear

Minor comments:
- in Algorithm 1 there is a condition given that the loss is smaller than zero but the loss is always positive?
- in the footnote the authors seem to suggest that they don't use the indicator function at all (gradient is set to zero everywhere) for the optimization but then it should also not be included in the objective - this kind of presentation is misleading
- the $\ell_1$-attacks in the paper seem no longer SOTA. I suggest to use instead
  [S1] Croce, Francesco and Matthias Hein. “Mind the box: l1-APGD for sparse adversarial attacks on image classifiers.” International Conference on Machine Learning
 and
 Brendel, W., Rauber, J., K uemmerer, M., Ustyuzhaninov, I, Bethge, M. Accurate, reliable and fast robustness evaluation. NeurIPS, 2019.
- it should be clarified that the $\ell_0$-threat model is channel-based and not pixel-based

Summary:
The discussion about the usefulness of the $\ell_0$-threat model played no role in the decision but compared to previous work, this paper only considers $\ell_0$-attacks so the scope is a bit limited. Given the large changes of the paper and the method as well as the missing comparisons, this paper is not ready yet for publication. However, I encourage the authors to finish the comparisons in a fair way and the resubmit their work.

**Justification For Why Not Higher Score:**

see discussion above

**Justification For Why Not Lower Score:**

N/A

---

### Decision · Program_Chairs · 2024-01-16

Reject